# Numerical Simulation of Shallow Geothermal Field in Operating of a Ground Source Heat Pump System—A Case Study in Nan Cha Village, Ping Gu District, Beijing

**Yaobin Zhang [1], Jia Zheng [2], Aihua Liu [2], Qiulan Zhang [1,*], Jingli Shao [1] and Yali Cui [1]**

[1]   School of Water Resources and Environment, China University of Geosciences (Beijing),
     Beijing 100083, China; zhangyaobinzybzyb@163.com (Y.Z.); jshao@cugb.edu.cn (J.S.);
     cuiyl@cugb.edu.cn (Y.C.)
[2]   Beijing Institute of Geothermal Research, Beijing 102218, China; zhengjiaspring@126.com (J.Z.);
     liuaihua0129@163.com (A.L.)
*   Correspondence: qlzhang919@cugb.edu.cn

**Abstract:** The inefficient use of single energy and cold accumulation in the shallow geothermal field seriously affect the efficient operation of the ground source heat pump system (GSHPS). The operation of solar-assisted GSHPS can effectively solve the above problems. In this paper, a shallow geothermal utilization project in Nan cha Village, Ping Gu District of Beijing, is chosen as the study area. A three-dimensional numerical model of groundwater flow and heat transfer considering ambient temperature and backfill materials is established, and the level of model integration and validation are novel features of this paper. The thermal response test data in summer and winter conditions are used to validate the model. The results show that increasing hydraulic gradient has a positive impact on the heat exchange. The mixture of sand and barite powder is recognized as a more efficient and economical backfill material. The changes of thermal influence radius, heat balance, and shallow geothermal field are simulated and analyzed by three schemes. It is demonstrated that the thermal influence radius is 5 m, 3.9 m and 3.9 m for Scheme 1, Scheme 2 and Scheme 3, respectively. The ground temperature is always lower than the initial formation temperature in Scheme 1 and Scheme 2; however, under Scheme 3 it is higher than the initial values. The closer the hole wall is, the larger the difference between the initial formation temperature and the ground temperature, and vice versa. The thermal equilibrium of Scheme 1, Scheme 2 and Scheme 3 is $-728 \times 106$ KJ, $-269 \times 106$ KJ and $+514 \times 106$ KJ. Through comprehensive analysis of the above three factors, Scheme 3 is regarded as the most reasonable Scheme for a solar system to assist GSHPS.

**Keywords:** shallow geothermal field; thermal equilibrium; influence radius of heat

## 1. Introduction

In recent years, the use of fossil energy has brought great pressure on the environment, so exploiting environmentally friendly energy becomes increasing imperative. Generally, renewable energy involves solar, wind and geothermal energy [1–3]. Shallow geothermal energy with the advantages of extensive distribution, huge reserves, mature mining technology and sustainability, is widely utilized in China, North America and Europe [1,4–8]. It can be demonstrated that the globally installed capacity is approximately 50,258 MWt and America is the predominant country in geothermal applications [9–13]. Furthermore, China's geothermal utilization rate is rising gradually [13].

Shallow geothermal energy, which refers to the resources stored in soil and groundwater within 200 m below the ground surface, can be exploited by extracting groundwater and utilizing a borehole heat exchanger. In a closed-loop ground source heat pump system, the fluid in the buried pipe does not directly contact the aquifer, rock and soil, but only removes heat or extracts heat through fluid circulation. In an open-loop groundwater heat pump system (GWHPS), groundwater is extracted for heat exchange, and then in principle groundwater is injected into the aquifer [14,15]. The recharge of groundwater is difficult and groundwater exploitation has a negative influence on the quality of groundwater. Changes in groundwater temperature can affect the redox reaction process and the survival of microbial communities in the aquifer. As a consequence, the sustainability of GWHPS may be influenced by a microbially mediated clogging process through biogeochemical reactions [16–21], so the borehole heat exchanger is the predominant way to make full use of shallow geothermal energy. However, in the northern mountainous areas of China, heating demand is greater than cooling demand, which causes the operating efficiency of the GSHPS to be low. The reason is that the ground temperature cannot be restored to the initial value before the beginning of each heating season. This causes the temperature of buried surrounding pipe to fall gradually. The shallow geothermal field is under a negative equilibrium state for a long time. When the GSHPS is used in succession, energy efficiency reduces year by year, while in these areas solar energy resources are relatively abundant, and this problem can be solved by the operation of solar-assisted GSHPS [22]. The principle of this method is that the solar energy system is used to supplement the shallow geothermal field to accelerate its the recovery in non-heating seasons. Meanwhile, a solar energy system can directly supply buildings when its fluid temperature is higher than the set temperature. In this way, it can improve energy efficiency. Shallow geothermal energy is mainly utilized through heat transfer in pipe fluid flow, backfill material and adjacent formation; the analysis methods of buried pipe heat transfer include the theoretical method, analytical method and numerical method [23]. The line source model and the cylinder source model are two basic examples of a theoretical method. The theoretical model has improved gradually, though the analytical method has advantages in calculation efficiency and accuracy. In general, an actual GSHPS mainly consists of an underground buried pipe group, heat pump system and the building units [23]. These factors affect the heat transfer process and system operating efficiency, such as the design of buried pipes, backfill materials and groundwater flow. Groundwater flow can improve the performance of the buried pipe in the process of heat transfer and should not be neglected [24–26]. The backfill material is situated between the U-shaped pipe and the borehole wall, which is used to enhance heat exchange between the buried pipe and the surrounding formation and can effectively prevent block vertical seepage in different aquifers [27,28]. In this way, it can reduce the drilling depth and the number of boreholes, thereby reducing economic costs [29–31]. Therefore, it is easier to use numerical methods to consider these factors in order to analyze the short-term and long-term response of GSHPS [32]. Numerical models of borehole heat exchangers (BHES) include the 2D heat transfer model and the 3D heat transfer model. The 2D heat transfer model of BHES includes the equivalent diameter method, EWS model, MISOS model, capacity resistance model, thermal resistance and capacity models and the composite medium infinite line source model [33]. The 3D heat transfer model of BHES includes FEM, FVM and FDM. It can be concluded that the 3D model has more obvious advantages in simulation accuracy and applicable scope than the 2D model [33,34]. The use of numerical simulation techniques for the shallow geothermal field can reveal the heat transfer process of buried pipes, the quantitative analysis of thermal equilibrium, and provide a rational basis for the utilization of shallow geothermal energy.

Many domestic and foreign scholars have conducted a large number of studies on the combined operation of solar-ground source heat pumps with a range of numerical simulation software such as TRNSYS [35], FLUENT [36], TOUGH2 [37,38], FEFLOW [39], OPENGEOSYS [40] and COMSOL [41]. The software can simulate the heat transfer process of buried pipes. FEFLOW and TOUGH2 are coupled with a groundwater flow package and a heat transfer package. Both are equivalent to the grid unit in simulating buried pipes. The effect of thermal dispersity on buried pipe heat transfer can be

studied and is more suitable for the simulation of groundwater source heat pump systems. TRNSYS is a modular simulation software that simulates the system's energy efficiency by using different modules. FLUENT is a special software for fluid flow but is not perfect in multi-physics coupling. It is difficult for TRNSYS and FLUENT software to perform preprocessing and post-processing simulation for specific systems [42]. OPENGEOSYS (OGS), which is based on the C++ platform, is an object-oriented free and open source program tool. The basic principle of OGS is the finite element method, and OGS can simulate single or combined thermal–hydraulics–mechanics–chemical processes in porous media and fractured media. It has huge potential especially in the geothermal field. COMSOL integrates the physical fields related to groundwater flow, heat transfer and fluid flow. This software has been widely used in geothermal, material, and chemical fields.

Zongwei Song et al. [43] proposed that the subsoil temperature was relatively low in the cold regions of China, where only GSHPS was used for heating. It was necessary to increase the number of heat exchange boreholes and the power of heat pumps, thereby increasing economic costs and causing negative impact on shallow geothermal fields. Finally, it was proposed that the combined operation of a solar-ground source heat pump could effectively solve the problem of an unbalanced geothermal field. Xinbo Lei et al. [41] used the finite element software COMSOL to establish a three-dimensional thermal infiltration coupling model and studied the effects of groundwater flow and buried pipe interaction on heat transfer efficiency and geothermal field. Moreover, the impacts of hydraulic gradient and permeability coefficient on the shallow geothermal field around buried pipes were analyzed emphatically. Chao Li et al. [44] utilized thermal response experiments and numerical simulations to analyze a geothermal field in Xi'an and evaluated the heat transfer capacity of buried pipe in the study area. Furthermore, the effect of water flow rate in tubes and system running time on heat transfer was analyzed. Min Li and Lai [42] reviewed different analytical models in the heat transfer process of vertical heat exchangers and evaluated the advantages and disadvantages of in-situ experiments, sandbox experiments, and numerical simulations on time and space scales. Finally, it was suggested that the problem lay in the heat transfer process of vertical buried pipes. Foreign researchers have also carried out many investigations on the operation of GSHPS. They studied the effects of spiral buried pipes and nanofluids on heat transfer, and also conducted simulations of solar-assisted GSHPS operations. It was demonstrated that solar energy could effectively compensate for the shallow geothermal field and improve the system's operating efficiency and economic benefits [45–49]. Han and Yu [50] used the monitoring data of a site in the United States to conduct a parameter sensitivity analysis based on a three-dimensional finite element model, and studied the impact of fluid flow velocity, thermal conductivity, specific heat capacity of the rock and soil, the system operating mode, and groundwater on heat pump efficiency. Aramzabal et al. [23] obtained the thermal conductivity of the formation at different depths through parameter inversion based on the experimental data of stable heat flow tests. Meanwhile, the existence of groundwater flow was determined by the curve of thermal conductivity with depth. However, most of the current studies focus on the energy efficiency and performance of GSHPS. The analysis of shallow geothermal field in terms of thermal influence radius, ground temperature and thermal equilibrium is relatively limited to numerical simulation considering groundwater flow, backfill material and ambient temperature simultaneously. In this work, the combination of solar energy system and ground source heat pump system is reflected mainly in the inlet temperature of the buried pipe. The variation of inlet temperature depends on the operation of a practical solar energy system, so the optical physics field is not added to the model.

In this paper, a three-dimensional model of fluid flow combined with thermal transport considering groundwater, backfill materials and ambient temperature conditions is established to simulate the heat transfer performance. The numerical model is validated by the thermal response test of two heat exchange boreholes and studies the impact of hydraulic gradient on the heat transfer of buried pipes. The effect of three different backfill materials, sand, cement mortar and the mixture of sand and barite powder, on the outlet temperature of buried pipe is compared. Furthermore, the horizontal and vertical geothermal distributions are analyzed. A comprehensive analysis of the thermal interaction radius, ground temperature, and thermal equilibrium is conducted to seek a reasonable operation scenario.

## 2. Materials and Methods

### 2.1. Geological Profile of the Study Area

The study area is located in Nan cha Village, Ping Gu District, Beijing (see Figure 1). This mountainous area is of significance for a project that intends to study the heating mode of shallow geothermal energy in the mountainous area of Beijing. According to the C1 borehole, the stratum from 0 to 15 m consists of Quaternary alluvial deposits, and the lithology is mainly cohesive soil, gravel, and their mixture. The stratum below 15 m is bedrock, which mainly includes siltstone, mudstone, argillaceous siltstone and quartz sandstone. The thickness of the bedrock is significantly larger than the Quaternary alluvial deposits. Groundwater flow condition is poor, and the water depth is 75 m. The ground temperature gradient of the study area is not obvious.

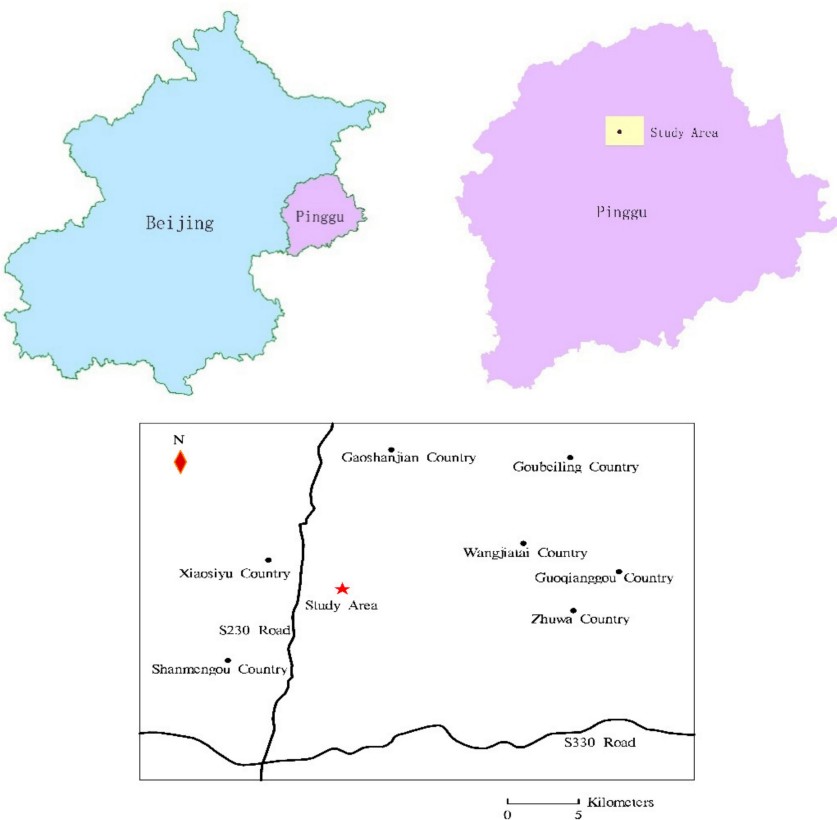

**Figure 1.** Location of the Study Area.

### 2.2. Heat Transfer Experiment

A total of 17 boreholes are designed in the study area (see Figure 2) with depth of 102 m and diameter of 152 mm. Among them is a double U-shaped channel with PE inserts in every vertical borehole. K1 to K12 boreholes are used as the borehole of heat exchange. M1 and M2 boreholes are used as the monitoring boreholes of heat exchange and temperature sensors are designed within the tube. The type of temperature sensor is MPT-RTU and its accuracy is ±0.5 °C. F1 and F2 boreholes are used as the effect monitoring boreholes of heat exchange, and temperature sensors are designed within the tube. C1 borehole as the exploration borehole is more than 10 m away from the borehole distribution area and temperature sensors are designed in the tube. The C1 borehole is regarded as constant temperature monitoring borehole. The K4 heat exchange borehole is backfilled with a mixture of sand and barite powder, and the K3 heat exchange borehole is backfilled with cement mortar. The remaining heat exchange boreholes are backfilled with sand. The K2, K3 and K4 heat exchange boreholes are equipped with temperature sensors at the inlet and outlet of the borehole, and a flowmeter is installed at the inlet to observe the real data of the system operation. The type

of flowmeter is LDBE, its accuracy is ±0.1%. F2, and C1 and M1 heat transfer boreholes have been resistivity logged and temperature measured.

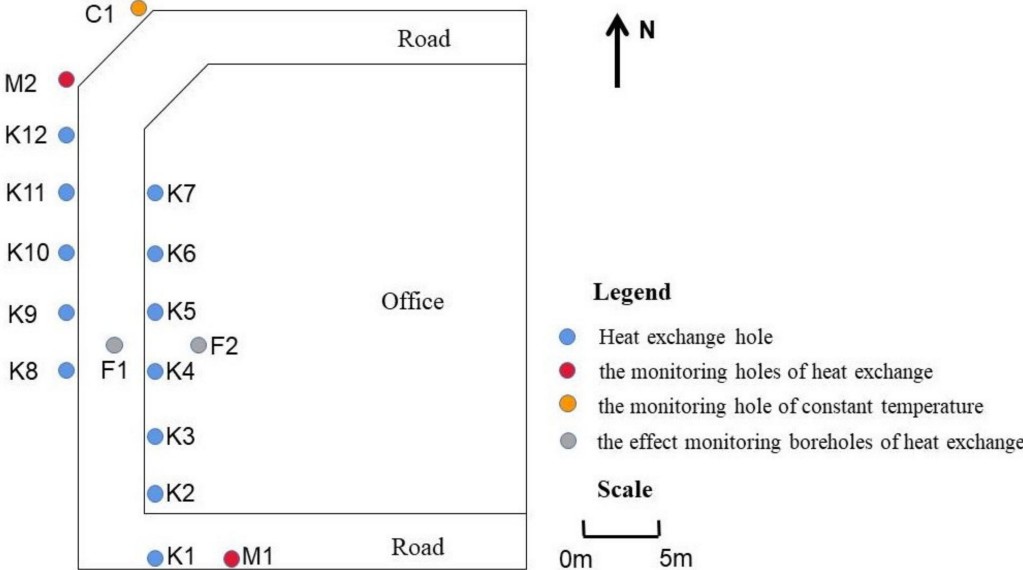

**Figure 2.** Schematic of heat exchange borehole distribution in the study area.

The thermal response tests [51,52] including the constant temperature condition and the constant power condition are carried out on the K2 borehole (see Figure 3), K3 borehole (see Figure 4) and K4 (see Figure 5) borehole respectively by precisely utilizing the GH-12WT09 cabinet type tester (see Figure 6). The summer and winter working condition experiments are tested on K3 borehole and K4 borehole. Only the summer constant temperature condition experiment is carried out on K2 borehole. Experiments with stable heat flow are conducted at 6 KW and 9 KW for three boreholes, respectively. The flow rate in the U-shaped tube is 1.5 m³/h and the results show that average thermal conductivity is 2.83 W/(m·K), 2.59 W/(m·K), 2.69 W/(m·K), respectively, for K2 borehole, K3 borehole and K4 borehole. The mean value is 2.70 W/(m·K). The inlet temperature of buried pipe in summer working conditions is 35 °C and the inlet temperature of buried pipe in winter working conditions is 5 °C. The flow rate in the U-shaped tube is 1.5 m³/h. The test results show that the heat exchange capacity is 74.52 W/m, 69.56 W/m, and 72.52 W/m, respectively, in summer conditions for K2 borehole, K3 borehole and K4 borehole. The mean value is 72.20 W/m. The heat exchange capacity is 44.86 W/m and 45.67 W/m, respectively, for K3 borehole and K4 borehole in winter conditions. The mean value is 45.27 W/m.

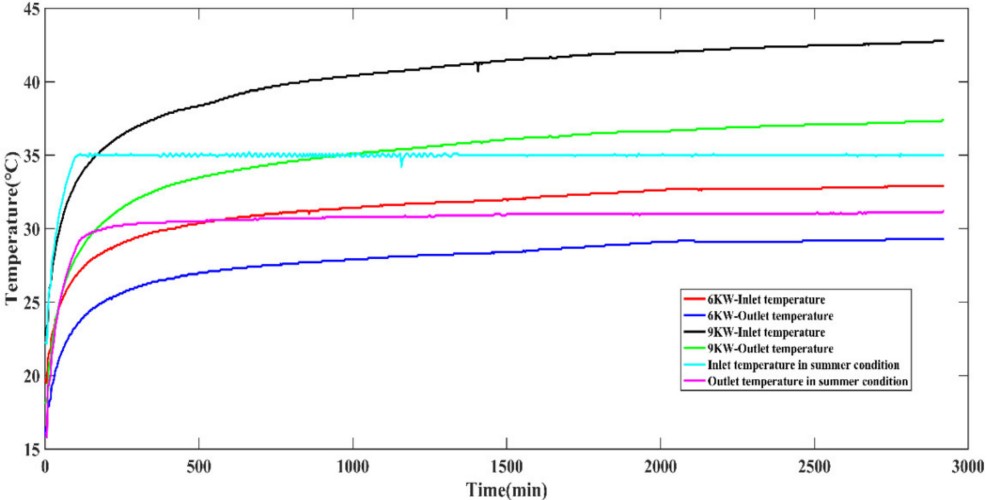

**Figure 3.** The results of thermal response test for K2 borehole.

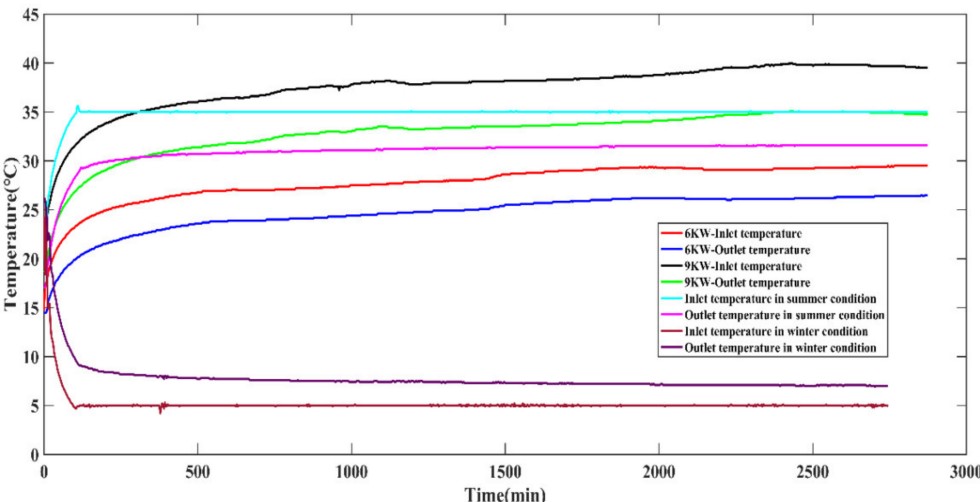

**Figure 4.** The results of thermal response test for K3 borehole.

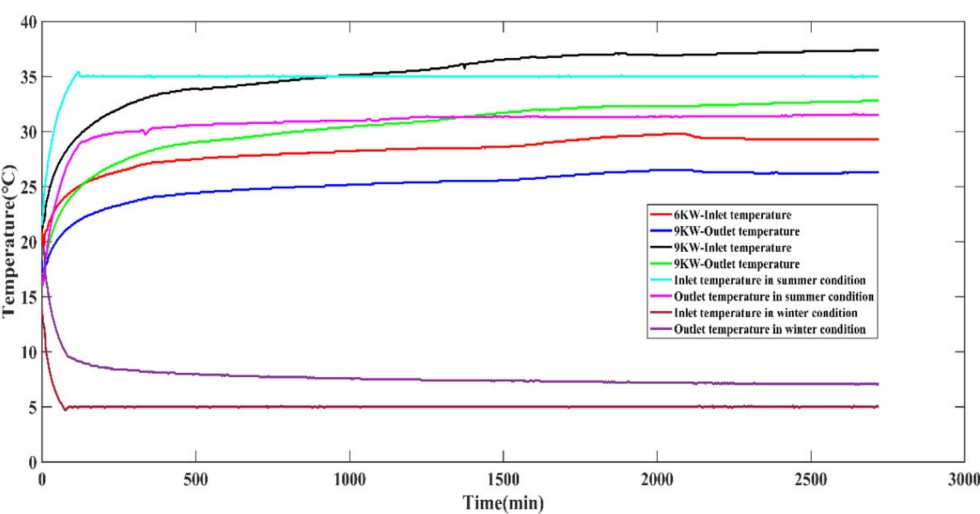

**Figure 5.** The results of thermal response test for K4 borehole.

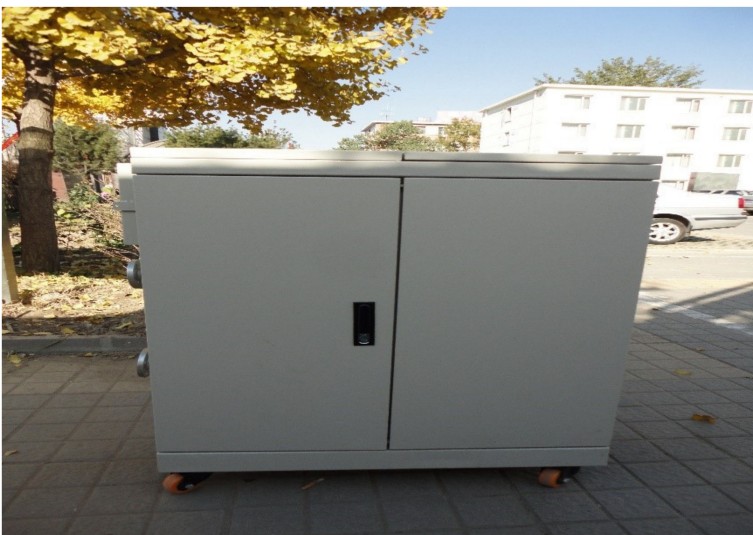

**Figure 6.** Diagram of GH-12WT09 cabinet type tester. Its power source is 3 Ph 380 V. The maximum heating power of tester is 12 KW. The maximum flow of water pump is 4 m$^3$/h. The accuracy of each part of the tester meets requirements.

A sample was taken in C1 borehole for every 10 m during the drilling, and each sample is 20 cm in length. Thermal properties including thermal conductivity, thermal diffusion coefficient and specific heat capacity are obtained by the HotDisk equipment [53]. Parameters such as density and porosity are obtained by empirical value in a similar area. The groundwater in the study area flows from south to north with an approximate hydraulic gradient of 5/1000 obtained by combining well data and water level measurement.

## 3. Numerical Simulation

### 3.1. Mathematical Model

There are three kinds of physical field involved in the study area; the physical fields and governing equations are as follows:

(1) The governing equation of heat transfer in non-isothermal pipe flow is

$$\rho A C_p \frac{\partial T}{\partial t} + \rho A C_p u e_t \cdot \nabla_t T = \nabla_t \cdot (A k \nabla_t T) + \frac{1}{2} f_D \frac{\rho A}{d_h} |u| u^2 + Q + Q_{wall} \tag{1}$$

where $\rho$ is the fluid density in the pipe, kg/m$^3$; $A$ is the pipe cross-sectional area, m$^2$; $C_p$ is the isobaric heat capacity of fluid, J/(kg·K); $u$ is the tangential velocity of the circulation fluid, m/s; $f_D$ is the friction coefficient; $d_h$ is the mean hydraulic diameter, m; $Q$ is the heat source, W/m$^3$; $T$ is the temperature of the fluid, K; $Q_{wall}$ is the heat exchange on the pipe wall. The calculation formula is as follows [54]:

$$Q_{wall} = (hz)_{eff} \left( T_{ext} - T_f \right) \tag{2}$$

where $(hz)_{eff}$ is the effective, total thermal resistance of the pipe wall, W/(m·K), which includes the thermal resistance of the pipe and the thermal resistances of the inner and outer pipe wall with the convection layer; $T_{ext}$ is the external temperature outside the pipe, K; and $T_f$ is the fluid temperature inside the pipe, K [40].

(2) The governing equations of heat transfer in porous media are expressed as

$$(\rho C_P)_{eff} \frac{\partial T}{\partial t} + \rho C_P u \cdot \nabla T + \nabla \cdot q = Q \tag{3}$$

$$q = -k_{eff} \nabla T \tag{4}$$

where $\rho$ is the fluid density, kg/m$^3$; $C_P$ is the isobaric heat capacity of fluid, J/(kg·K); q is the heat flux, W/m$^2$; u is the groundwater flow velocity, m/s; $k_{eff}$ is the equivalent thermal conductivity, W/(m·K); $k_{eff} = (1-\varphi)k_s + \varphi k_l$, $k_s$ is the thermal conductivity of solids and $k_l$ is the thermal conductivity of liquid, W/(m·K); $\varphi$ is porosity, %; Q is the heat source or heat sink, W/m$^3$.

(3) The governing equations of groundwater flow are

$$\frac{\partial}{\partial t} \left( E_p \rho \right) + \nabla (\rho v) = Q_m \tag{5}$$

$$v = -\frac{k}{u} \nabla p \tag{6}$$

where $\rho$ is the fluid density, kg/m$^3$; $E_p$ is porosity, %; k denotes the permeability of the porous medium, m$^2$; $p$ is the pressure, Pa; $Q_m$ is a mass source term, kg/(m$^3$·s); $v$ is the Darcy velocity, m/s; $u$ is the dynamic viscosity of the fluid, Pa·s. Porosity is defined as the fraction of the control volume that is occupied by pores. Thus, the porosity can vary from zero for pure solid regions to unity for domains of free flow.

In this study, a numerical FEM software COMSOL is used to establish a numerical model of the study area. The model contains the above mentioned three fields. Experimental data of temperature at the outlet of the buried pipe in the thermal response test is used to validate the model.

During the model establishment process, due to the limitation of the conditions, the following assumptions are considered: (1) Soil and backfill materials are isotropic, and their thermal properties are constant. (2) Ignore the effects of water movement in soil. (3) Ignore the heat transfer around the U-shaped pipe in vertical direction - the heat is only transmitted in the horizontal direction. (4) Double U-shaped pipe is equivalent to single U-shaped pipe and the diameter of single U-shaped pipe is $\sqrt{2}$ times that of Double U-shaped pipe. (5) Impact of surface frozen soil layer is not considered.

*3.2. Geometric Model and Parameters*

In order to accurately simulate the actual formation and heat transfer of the ground buried pipe, the model is generalized to 19 layers with a length, width, and height of 100 m × 100 m × 120 m (see Figure 7). The nest of tubes consists of 17 buried pipes. The U-shaped pipe is buried at a depth of 100 m and buried pipes are arranged at a distance of 5 m in addition to C1 borehole. The model parameters mainly include buried pipe parameters and fluid parameters (as shown in Table 1), and thermal properties of the formation, as shown in Table 2.

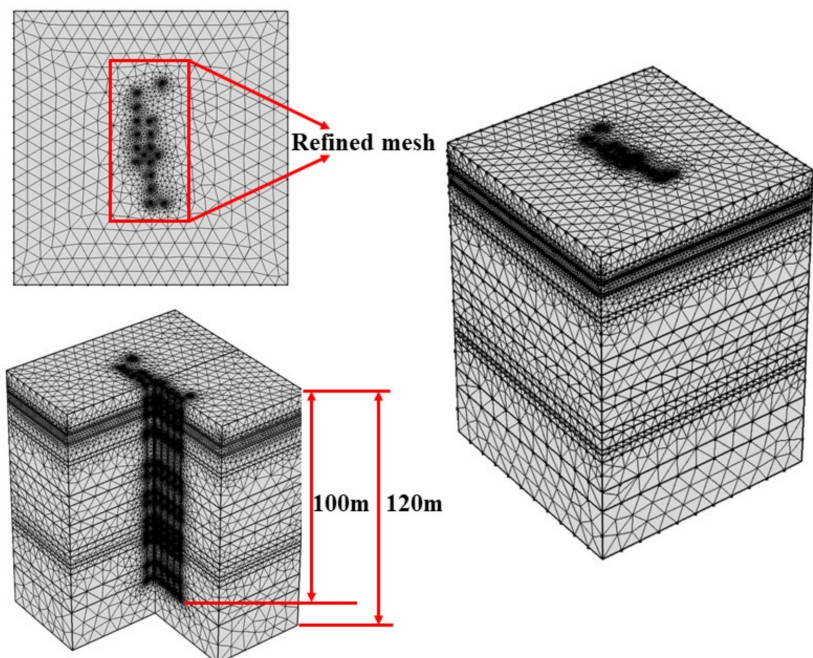

**Figure 7.** Schematic of the mesh.

**Table 1.** The parameters of borehole and buried pipe.

| Items | Unit | Number |
| --- | --- | --- |
| The depth of the hole | m | 102 |
| The depth of buried pipe | m | 100 |
| The diameter of the drilling | mm | 152 |
| Outer diameter of buried pipe | mm | 32 |
| Inner diameter of buried pipe | mm | 26 |
| Flow rate of circulating fluid | m$^3$/h | 1.50 |
| Thermal conductivity of U-tube | W/(m·K) | 0.42 |
| Thermal conductivity of sand and barite powder | W/(m·K) | 2.20 |
| Thermal conductivity of cement mortar | W/(m·K) | 1.76 |
| Thermal conductivity of sand | W/(m·K) | 1.10 |

**Table 2.** Characteristics of each layer.

| Layer | Depth m | Lithology | Thermal Conductivity W/(m·K) | Specific Heat Capacity M/(m³·K) | Thermal Diffusion Coefficient mm²/s | Density kg/m³ | Porosity | Penetration Rate m² |
|-------|---------|-----------|------------------------------|---------------------------------|--------------------------------------|---------------|----------|---------------------|
| 1 | 0–6.8 | Clayey soil | 1.57 | 2.60 | 0.66 | 1900 | 0.60 | $5 \times 10^{-6}$ |
| 2 | 6.8–9.4 | Clayey soil with detritus | 2.1 | 2.03 | 0.72 | 1900 | 0.55 | $5 \times 10^{-6}$ |
| 3 | 9.4–10.3 | Gravel | 2.1 | 2.03 | 0.72 | 1900 | 0.50 | $5 \times 10^{-6}$ |
| 4 | 10.3–14 | Clayey soil with detritus | 2.1 | 2.03 | 0.72 | 1900 | 0.55 | $5 \times 10^{-6}$ |
| 5 | 14–15 | Gravel | 2.1 | 2.03 | 0.72 | 1900 | 0.50 | $5 \times 10^{-6}$ |
| 6 | 15–23 | Siltite | 2.07 | 1.41 | 1.48 | 2200 | 0.02 | $5 \times 10^{-10}$ |
| 7 | 23–25 | Mud rock | 1.59 | 1.69 | 0.94 | 2200 | 0.06 | $5 \times 10^{-10}$ |
| 8 | 25–40 | Siltite | 2.07 | 1.41 | 1.48 | 2200 | 0.02 | $5 \times 10^{-10}$ |
| 9 | 40–45 | Argillaceous siltite | 1.59 | 1.43 | 0.75 | 2200 | 0.02 | $5 \times 10^{-10}$ |
| 10 | 45–54.8 | Siltite and sand rock | 2.07 | 1.86 | 1.11 | 2200 | 0.02 | $5 \times 10^{-10}$ |
| 11 | 54.8–59.4 | Quartz sand rock | 1.95 | 1.67 | 1.17 | 2500 | 0.01 | $6 \times 10^{-10}$ |
| 12 | 59.4–68 | Siltite | 2.07 | 2.36 | 1.32 | 2200 | 0.02 | $5 \times 10^{-10}$ |
| 13 | 68–71 | Quartz sand rock | 3.95 | 1.85 | 2.13 | 2500 | 0.01 | $6 \times 10^{-10}$ |
| 14 | 71–75 | Argillaceous siltite | 1.59 | 1.43 | 0.75 | 2200 | 0.02 | $5 \times 10^{-10}$ |
| 15 | 75–77 | Siltite | 1.88 | 1.53 | 1.23 | 2200 | 0.02 | $5 \times 10^{-10}$ |
| 16 | 77–80 | Argillaceous siltite | 1.59 | 1.43 | 0.75 | 2200 | 0.02 | $5 \times 10^{-10}$ |
| 17 | 80–92 | Siltite | 1.88 | 1.53 | 1.23 | 2200 | 0.02 | $5 \times 10^{-10}$ |
| 18 | 92–100 | Quartz sand rock | 2.5 | 1.85 | 2.13 | 2500 | 0.01 | $6 \times 10^{-10}$ |
| 19 | 100–120 | Quartz sand rock | 2.5 | 1.85 | 2.13 | 2500 | 0.01 | $5 \times 10^{-10}$ |

### 3.3. Initial and Boundary Conditions

The initial conditions are as follows: groundwater flow in the study area from south to north and approximate hydraulic gradient is 5/1000. The initial shallow geothermal field is shown in Table 3.

**Table 3.** Initial ground temperature.

| Depth (m) | 10 | 20 | 30 | 40 | 50 | 60 | 70 | 80 | 90 | 100 | 110 | 120 |
|-----------|----|----|----|----|----|----|----|----|----|-----|-----|-----|
| Temperature (°C) | 12.83 | 12.68 | 12.63 | 12.60 | 12.80 | 12.58 | 12.80 | 12.88 | 12.83 | 12.93 | 13.00 | 13.00 |

The boundary conditions are as follows: set the model surroundings to a constant temperature boundary according to the physics of heat transfer in porous media. The bottom boundary is set to constant temperature at a value of 13 °C and the top boundary is defined as ambient temperature. According to the physics of heat transfer non-isothermal pipe flow, set the inlet temperature and inlet flow rate at the entrance of the buried pipe. Set the U-shaped pipe outlet to constant pressure boundary. According to the physics of Darcy's law, the surroundings of the model are set as the hydraulic head boundary and the no-flow boundary, respectively.

### 3.4. Discretization of Time and Space

The model takes into account the balance between calculation accuracy and calculation efficiency. It is discretized by refinement mesh, which is automatically refined around the buried pipe, while the size of mesh far away from the buried pipe is bigger. The biggest unit size is 9.6 m and the smallest unit size is 1.2 m. The model has in total 888,452 units, as shown in Figure 3. The simulation period of the model is ten years, and the time step is one day.

### 3.5. Model Verification

It can be seen that the inlet temperature is set to 35 °C in summer conditions and to 5 °C in winter conditions for K3 borehole. The fitting curves of the measured and simulated outlet temperature of buried pipe can be found in Figure 8. The temperature tends to be stable when the thermal response test experiment is conducted for 180 min and 342 min in winter and summer working conditions, respectively. Finally, the simulated outlet temperature is higher than the measured outlet temperature in summer working conditions. As a result, the absolute error value is 0.3 °C and correlation coefficient is 0.92. The simulated outlet temperature is higher than the measured outlet temperature in winter working conditions. As a consequence, the absolute error value is 0.6 °C and correlation coefficient is 0.92. It can be concluded that the outlet temperature calculated by the numerical model is basically consistent with the outlet temperature recorded by the tester.

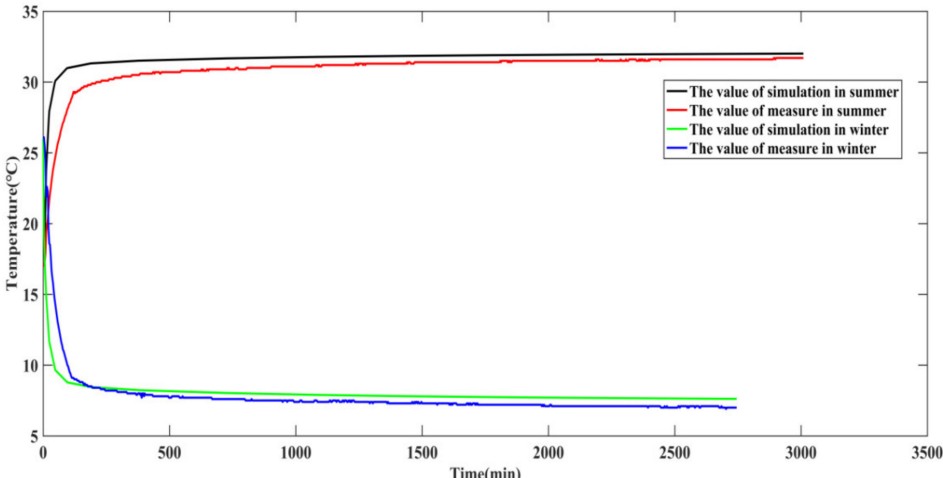

**Figure 8.** Thermal response test verification for K3 borehole; the horizontal axis represents the thermal response experiment time, and the vertical axis represents the outlet temperature of buried pipe.

Similarly, for K4 borehole, as shown in Figure 9, the temperature tends to be stable when the thermal response test experiment is conducted for 231 min and 369 min in winter and summer working conditions, respectively. Finally, the simulated outlet temperature is higher than the measured outlet temperature in summer working conditions. As a result, the absolute error value is 0.2 °C and correlation coefficient is 0.94. The simulated outlet temperature is higher than the measured outlet temperature in winter working conditions. As a consequence, the absolute error value is 0.8 °C and correlation coefficient is 0.99. It is shown that the simulation value agrees well with the observation value.

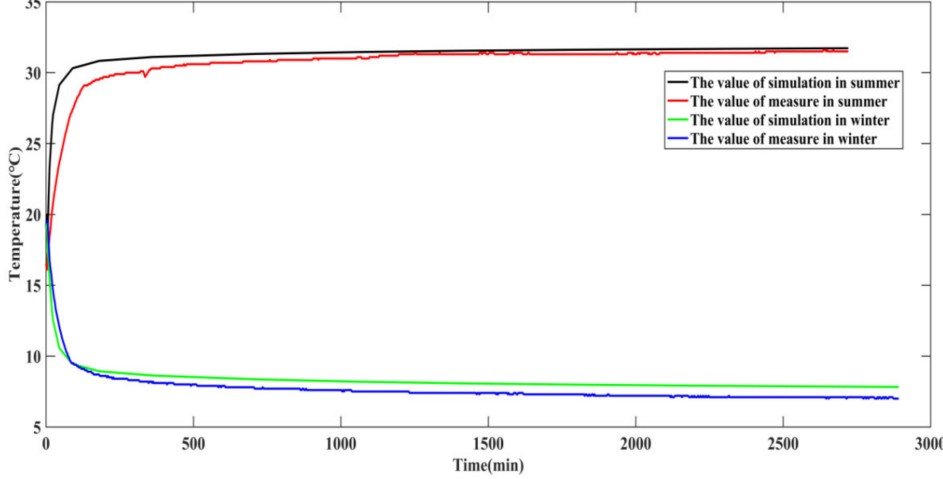

**Figure 9.** Thermal response test verification for K4 borehole; the horizontal axis represents the thermal response experiment time, and the vertical axis represents the outlet temperature of buried pipe.

## 4. Results and Discussion

### 4.1. Results Analysis

#### 4.1.1. Analysis of Thermal Equilibrium in the Validation Period

In the simulation of the summer working conditions of K3 and K4 holes, the volume of heat exchange in summer is 1,256,558.4 KJ and 1,162,411.2 KJ, respectively. The volume of heat exchange in winter is 739,503 KJ and 792,986.4 KJ, respectively. Obviously, the shallow geothermal field is under a positive equilibrium state and the value is 517,055.4 KJ and 369,424.8 KJ, respectively. Due to the difference in thermal response experiment time in winter and summer, there is a discrepancy in the volume of heat exchange. In summary, a positive equilibrium state is beneficial to the upcoming heating period.

#### 4.1.2. Analysis of the Influence of Hydraulic Gradient on Heat Exchange

In order to investigate the impact of hydraulic gradient on outlet temperature of the buried pipe, the relationship is obtained by simulating the thermal response test for summer working conditions under the premise that other parameters remain unchanged.

As the hydraulic gradient increases, the outlet temperature of buried pipe continues to decrease. But the rate of decrease continues to get smaller. Under a certain permeability coefficient, groundwater flow rate improves with the increment in the hydraulic gradient, which will make the effect of heat convection more obvious, so the heat exchange efficiency of buried pipe is improved and it is difficult for the phenomenon of cold and hot accumulation around the buried pipe to occur, as shown in Figure 10.

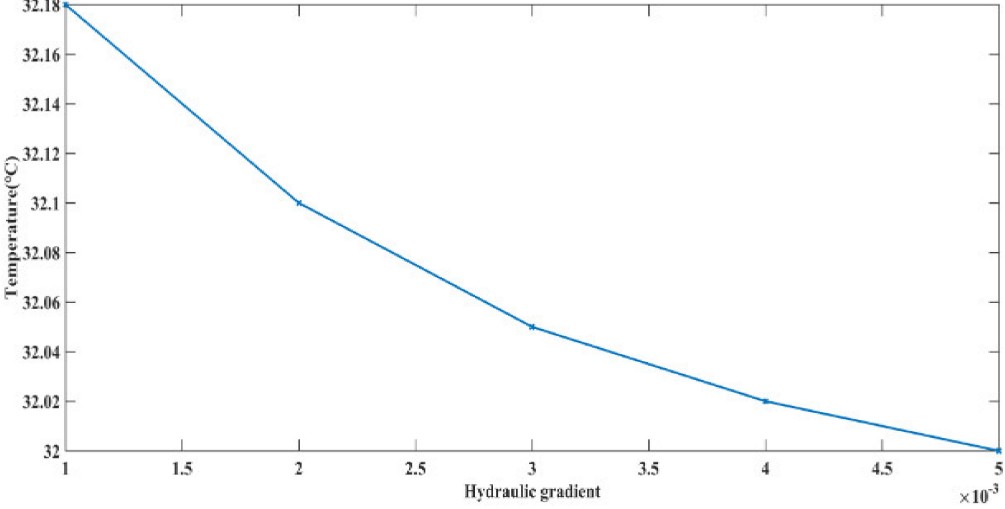

**Figure 10.** Effect of hydraulic gradient on outlet temperature of ground buried pipe.

#### 4.1.3. Analysis of the Influence of Backfill Material Outlet Temperature

The backfill material is part of the ground source heat pump system. Heat transfers between the buried pipes and adjacent formation through the backfill material, so the backfill material controls the thermal resistance of the pathway between the buried pipes and subsoil. The thermal conductivity of backfill material plays a significant role in the performance of GSHPS.

The backfill material for K2 borehole is sand and its thermal conductivity is 1.1 W/(m·K). The backfill material for K3 borehole is cement mortar and its thermal conductivity is 1.76 W/(m·K). The backfill material for K4 borehole is a mixture of sand and barite powder and its thermal conductivity is 2.2 W/(m·K). As shown in Figure 11, after approximately 170 min, the outlet temperature of buried pipe tends to be steady. It can be seen that the outlet temperature of the buried pipe for K2 borehole is the

highest in summer working conditions and the outlet temperature of the buried pipe for K4 borehole is the lowest in summer working conditions. Within a certain range, as the thermal conductivity of backfill material increases, the outlet temperature of buried pipe decreases gradually. It is concluded that backfill material with relatively high thermal conductivity facilitates heat exchange between buried pipe and surrounding formation. By comparison, the mixture of sand and barite powder is recognized as a more efficient and economical backfill material.

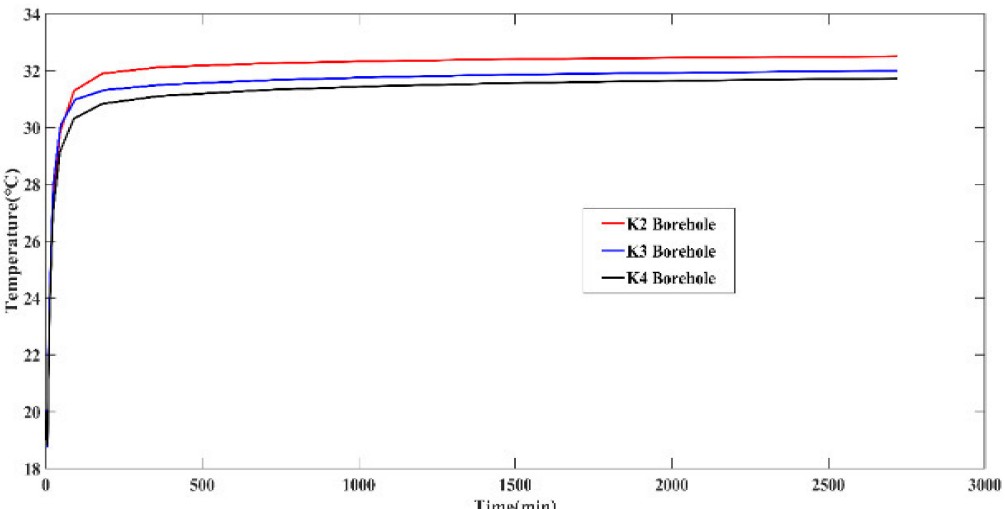

**Figure 11.** Effect of backfill material on outlet temperature of ground buried pipe.

*4.2. The Prediction Analysis of Different Scenarios*

4.2.1. Three Scenarios and Their Simulated Operating Conditions

In Scheme 1, the GSHPS operates continuously for 10 years with an inlet temperature of 7 °C and annually the system operates for five months in winter and seven months under closing state during the period of operation. Scheme 2 is similar to Scheme 1 and the only difference is that the inlet temperature of buried pipe is kept at 9.58 °C. In Scheme 3, the system operates continuously for 10 years and every year the system works five months in winter with an inlet temperature of 9.58 °C, one month in summer with an inlet temperature of 20 °C, and six months under closing state. In Scheme 3, the solar energy system is used in both winter and summer. In winter, the inlet temperature of the buried pipe is appropriately increased from 7.58 degrees to 9 degrees to reduce the occurrence of negative equilibrium of the shallow geothermal field. In summer, the ground source heat pump system is rarely used for cooling in mountainous areas. Therefore, in summer, a solar system with 20 degrees of water is used to supplement the heat of the shallow geothermal field and GSHPS to prepare for the next heating season. The setting of the solar system in the model is to increase the inlet temperature of the buried pipe, as shown in Table 4.

**Table 4.** Comparison of three scenarios.

| Scheme | Unit | Scenario 1 | Scenario 2 | Scenario 3 |
|---|---|---|---|---|
| Operation season | - | winter | winter | winter and summer |
| Operation period | month | 5 | 5 | 6 |
| Recovery time | month | 7 | 7 | 6 |
| Prediction time | year | 10 | 10 | 10 |
| With or without solar heating | - | without | with | with |
| Inlet temperature under heating condition | °C | 7 | 9.58 | 9.58 |
| Inlet temperature under cooling condition | °C | - | - | 20 |
| Fluid flow rate under operation condition | m$^3$/h | 1.5 | 1.5 | 1.5 |
| Fluid flow rate under closing condition | m$^3$/h | 0 | 0 | 0 |

#### 4.2.2. Analysis of Thermal Influence Radius

The thermal influence radius of the buried pipe is a significant design parameter of the ground source heat pump system. The heat influence radius determines the design spacing of the boreholes. Theoretically, the buried pipe is placed in semi-infinite rock and soil and the heat exchange process will affect the entire temperature distribution in the formation, but due to the slow thermal diffusion process the temperature attenuates as the distance increases during the thermal diffusion process, to a certain extent. The distance at this time can be used as heat influence radius. This study mainly uses the probe function in the software to compute the heat influence radius.

Under the prerequisite of constant flow in the pipe, the heating season is 150 days and the influence of the inlet temperature on heat impact radius is analyzed. The heat influence radius of Scheme 1 is 5 m, and the heat influence radius of Schemes 2 and 3 are both 3.9 m. The process of heat transfer among the buried pipes has been mutually affected, but the heat interference of Scheme 2 and Scheme 3 is smaller than Scheme 1.

#### 4.2.3. Analysis of Temperature Change Trend

(1)    Analysis of temperature changes at different positions from the buried pipe

Scheme 1 simulates the ground temperature changes of $r = 0.3$ m, $r = 0.5$ m, $r = 1$ m, $r = 2$ m, $r = 3$ m, $r = 4$ m, and $r = 5$ m during the ten year operation of the system (see Figure 12). It records a dramatic variation at $r = 0.3$ m from 12.59 °C in the first year to 10.83 °C in the tenth year, down 1.76 °C. The temperature value at $r = 0.5$ m falls from 12.59 °C in the first year to 10.85 °C in the tenth year, down 1.74 °C. The temperature value at $r = 1$ m decreases from 12.58 °C in the first year to 10.91 °C in the tenth year, down 1.67 °C. The temperature value at $r = 2$ m drops from 12.59 °C in the first year to 11.02 °C in the tenth year, down 1.57 °C. The temperature value at $r = 3$ m declines from 12.60 °C in the first year to 11.15 °C in the tenth year, down 1.45 °C. The temperature value at $r = 4$ m decreases from 12.60 °C in the first year to 11.28 °C in the tenth year, down 1.32 °C. The temperature value at $r = 5$ m sees a steady fall from 12.60 °C in the first year to 11.40 °C in the tenth year, down 1.20 °C.

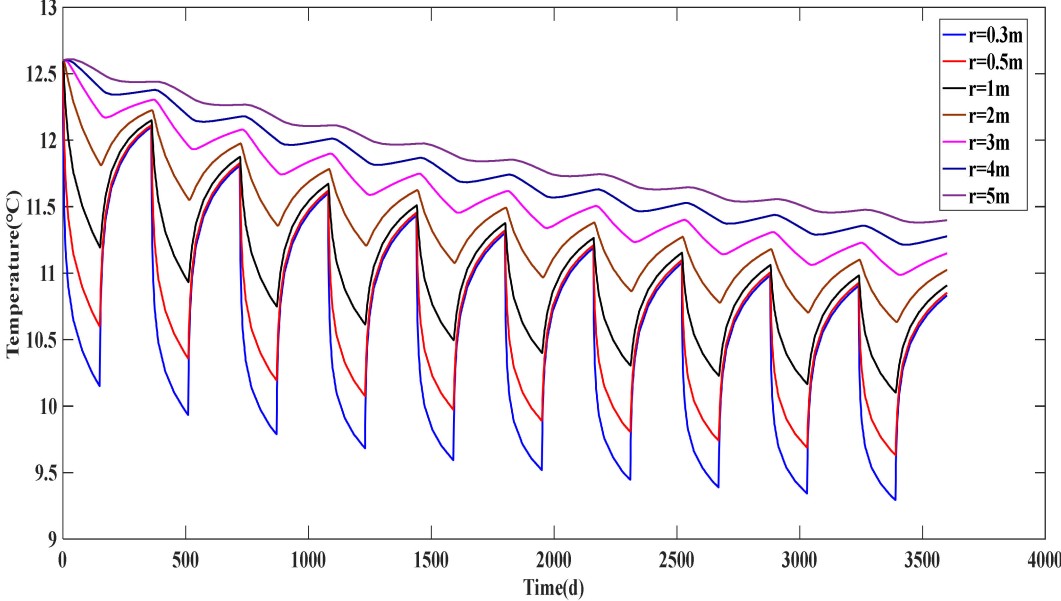

**Figure 12.** Temperature variation of K1 borehole at 60 m depth for ten years of continuous operation; *r* represents different position from the borehole.

As the operating time increases, the ground temperature tends to decrease, indicating that the ground temperature does not reach the initial value before each heating season, which causes the

negative influence on heat transfer. In conclusion, the closer the distance is, the larger the change is, and the longer the distance is, the smaller the change is. The two have a negative correlation.

(2)    Analysis of temperature variation at different depths

The thermal conductivity of the formation at the depth of 30 m and 50 m is the same, but the temperature of the ground is different. Therefore, we can see from Figure 13 that, in winter working conditions, the temperature decreases at a depth of 30 m by 2.51 °C in the period of heat transfer and increases by 1.90 °C in the period of recovery. The temperature decreases at a depth of 50 m by 2.68 °C in the period of heat transfer and increases by 2.01 °C in the period of recovery, so the buried pipe has higher efficiency of heat transfer at deeper depth. The ground temperature at 50 m and 70 m depths is the same, but the formation lithology at 70 m depth is quartz sandstone, which has a higher thermal conductivity; however, the temperature change at a depth of 70 m is smaller. As a consequence, the temperature decreases at a depth of 70 m by 2.36 °C in the period of heat transfer and increases by 1.79 °C in the period of recovery. Because the fluid in the tube has a smaller temperature difference between the formation, the heat exchange is slower, and the ground temperature is more likely to rise during the recovery period. It can be demonstrated that the change trend in ground temperature at different depths is similar, which is mainly affected by the thermal conductivity of the formation and the ground temperature value.

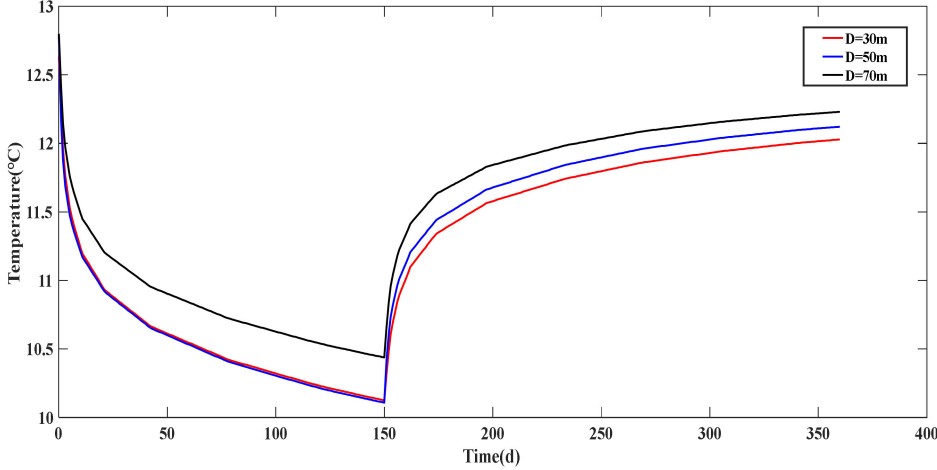

**Figure 13.** Ground temperature changes at different depth for K1 borehole.

4.2.4. Analysis of the Temperature Change Rule of the Buried Pipe Outlet

As shown in Table 5, the outlet temperature of buried pipe in Scheme 1 falls from 7.75 °C in the first year to 7.53 °C in the tenth year, down 0.22 °C. The outlet temperature of buried pipe in Scheme 2 sees a small drop from 9.98 °C in the first year to 9.85 °C in the tenth year, down 0.13 °C. The outlet temperature of buried pipe in Scheme 3 remains basically unchanged between 9.93 °C and 9.98 °C for ten years.

**Table 5.** Data of the outlet temperature of buried pipe (°C).

| Year | | 1 | 2 | 3 | 4 | 5 | 6 | 7 | 8 | 9 | 10 |
|---|---|---|---|---|---|---|---|---|---|---|---|
| | Scheme 1 | 7.75 | 7.69 | 7.65 | 7.63 | 7.61 | 7.59 | 7.57 | 7.55 | 7.54 | 7.53 |
| Outlet temperature | Scheme 2 | 9.98 | 9.95 | 9.93 | 9.92 | 9.90 | 9.89 | 9.88 | 9.87 | 9.86 | 9.85 |
| | Scheme 3 | 9.98 | 9.97 | 9.97 | 9.96 | 9.95 | 9.95 | 9.94 | 9.94 | 9.94 | 9.93 |

As the operating time increases, the outlet temperature continues to decrease, but the decreasing rate is slower and slower. Because the temperature of the fluid in the buried pipe and the ground

temperature value are obviously different during the first year of operation, the rate of heat transfer is fast, and the outlet temperature of buried pipe is relatively high. However, the ground temperature cannot be restored to the initial value during the natural recovery period, which causes the ground temperature value to continue to decrease. Again, the difference between the temperature of the fluid in the buried pipe and the ground temperature value becomes smaller and smaller, hence the amount of heat exchange continues to decrease. Therefore, the outlet temperature of buried pipe decreases gradually and the rate of decrease is slower and slower. It is concluded that the inlet temperature of buried pipe can be appropriately raised through the solar system under the premise of ensuring the heating load of the building, which is more advantageous to the natural restoration of the ground temperature field. Through the comparison of outlet temperature, Scheme 3 is more reasonable.

### 4.2.5. Analysis of Thermal Equilibrium

It is essential to conduct the analysis of thermal equilibrium in the study area. The thermal equilibrium term mainly includes heat transfer of the soil in winter, the variation of heat during the period of recovery, and heat exchange of the soil in summer. According to the calculation of thermal equilibrium by the thermal storage method from the Technical Regulations for Shallow Geothermal Energy Investigation and Evaluation, the equilibrium value of Scheme 1 is $-728 \times 106$ KJ, the equilibrium value of Scheme 2 is $-269 \times 106$ KJ, and the equilibrium value of Scheme 3 is $+514 \times 10^6$ KJ.

In terms of Scheme 1, there is a dramatic fall in the figure for the volume of heat exchange in winter from $271.2 \times 10^6$ KJ in the first year to $186.9 \times 10^6$ KJ in the tenth year, with the trend decreasing modestly from $180.4 \times 10^6$ KJ in the first year to $127.0 \times 10^6$ KJ in the tenth year for the volume of heat recovery. The figure remains relatively minor, between $-90.8 \times 10^6$ KJ and $-59.9 \times 10^6$ KJ for thermal equilibrium.

With the increment of operating time, the volume of heat exchange in winter and the volume of heat recovery continuously decreases and the rate of decrease is slower and slower for Scheme 1 (see Figure 14), but there is a singular point in the second year. This may be caused by numerical oscillation. The thermal equilibrium amount in Scheme 1 and Scheme 2 shows a negative state every year, illustrating that the amount of cold is constantly accumulating, and the heat exchange efficiency is decreasing. In Scheme 3, the thermal equilibrium presents a positive state every year. It is demonstrated that solar energy can promote the recovery of the shallow geothermal field. Meanwhile, the shallow geothermal field can store excess heat in the subsoil around the buried pipe and the heat transfer efficiency of buried pipes is improved in winter.

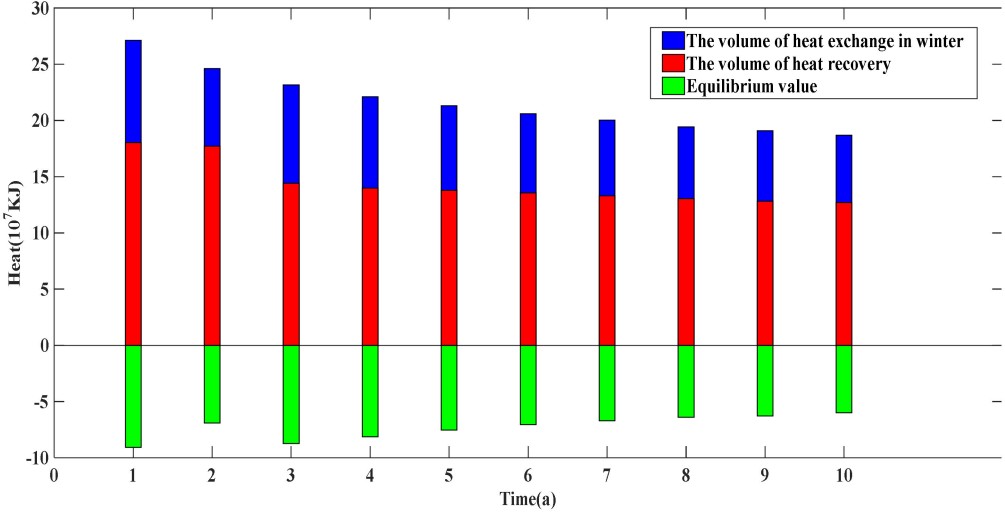

**Figure 14.** Thermal equilibrium in Scheme 1 for ten years of continuous operation.

#### 4.2.6. Preferred Solution

Through the comparison of the above aspects, the heat transfer efficiency of buried pipe is relatively high in Scheme 3. The GSHPS will consume a large amount of electrical energy and the economic benefits are poor in Schemes 1 and 2. By the simulation prediction for ten years, the system in Scheme 3 is more sustainable. Therefore, Scheme 3 is the best scheme.

### 5. Conclusions

In this paper, a three-dimensional numerical model of fluid flow combined with thermal transport considering groundwater flow, air temperature and backfill materials, is established. Three physical fields, coupled with heat transfer in porous media, non-isothermal pipe flow and Darcy's law, were used for long-term serial simulation predictions. The following conclusions can be obtained.

(1) The increase of hydraulic gradient has a positive impact on heat transfer. The outlet temperature of buried pipe decreases with the increase of the hydraulic gradient under summer working condition. It can be concluded that the increase of hydraulic gradient can improve the efficiency of heat exchange for GSHPS.

(2) The backfill material plays a significant role in the process of heat transfer. Within a certain range, as the thermal conductivity of backfill material increases, the outlet temperature of buried pipe decreases gradually and GSHPS shows an efficient performance. It is concluded that the mixture of sand and barite powder is recognized as a more efficient and economical backfill material.

(3) The variation of ground temperature at different depths is mainly affected by the thermal conductivity of the formation and initial value of ground temperature. In the horizontal direction, the closer to the buried pipe, the greater the difference between initial ground temperature and ground temperature at the end of the system, and the longer the distance is, the smaller the change. The two have a negative correlation.

(4) The heat interaction radius of Scheme 1 is 5 m, and the heat interaction radius of Scheme 2 and Scheme 3 are both 3.9 m. The heat transfer between different boreholes in the three operating modes mutually interferes, but the effect of heat interference in Schemes 2 and 3 is small.

(5) According to the change rule of the shallow geothermal field, the ground temperature in Scheme 1 and Scheme 2 continuously decreases with time and is lower than the initial value, which causes the volume of heat exchange to decrease gradually. However, the ground temperature before the beginning of each heating season in Scheme 3 is higher than its initial value, so the heat transfer efficiency of buried pipe is relatively high.

(6) Through the calculation of heat equilibrium, the thermal equilibrium of Scheme 1 is $-728 \times 106$ KJ, the thermal equilibrium of Scheme 2 is $-269 \times 106$ KJ, and the thermal equilibrium of Scheme 3 is $+514 \times 106$ KJ. This shows that cold accumulation occurs around the buried pipes in Scheme 1 and Scheme 2. Therefore, after comprehensively considering the heat interaction radius, ground temperature and thermal equilibrium, Scheme 3 is more reasonable.

(7) The semi-coupling model established in this study did not consider the optical radiation physical field. The solar system has not been characterized in detail, but parameters have been set through calculations and measured data. The uncertainty analysis of each parameter on the shallow geothermal field can be further studied.

**Author Contributions:** Formal analysis, Y.Z. and Q.Z.; investigation, J.Z. and A.L.; data curation, Y.Z., J.Z., A.L., Q.Z., J.S. and Y.C.; writing—original draft preparation, Y.Z.; writing—review and editing, Q.Z., J.S. and Y.C.; supervision, Q.Z., J.S. and Y.C. All authors have read and agreed to the published version of the manuscript.

**Funding:** This research was funded by the Research project on heating pattern of shallow geothermal energy in mountainous area of Beijing (No. PXM2019_158309_000008).

**Acknowledgments:** We sincerely thank the reviewers for useful comments and suggestions. Meanwhile, we appreciate Zilong Jia for the paper data from Beijing Institute of Geothermal Research.

**Conflicts of Interest:** The authors declare that we do not have any commercial or associative interest that represents a conflict of interest in connection with the work submitted.

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
