# Peer review of "Numerical Simulation of Shallow Geothermal Field in Operating of a Ground Source Heat Pump System—A Case Study in Nan Cha Village, Ping Gu District, Beijing"

_water, doi:10.3390/w12102938_

Round 1

Reviewer 2 Report

I have read with great interest the manuscript entitled "Numerical Simulation of Geothermal Field in Combined Operating of a Solar-ground Source Heat Pump---A case study in Nan Cha Village, Ping Gu District, Beijing". I think it is a very interesting and well-structured work. The modelling taks has been correctly carried out and I have not appreciated technical errors. So I recommend its publication in the Water Journal after minor revisions.

- A more extensive bibliographic analysis should be done in the introduction section. In many cases, important citations regarding the numerical modeling of shallow geothermal systems are missing. A very similar case of study is Zaragoza city where many numerical models of the geothermal systems installed in the Quaternary alluvial aquifer have been performed. For example:

García-Gil et al., 2020: Defining the exploitation patterns of
groundwater heat pump systems. Science of the total environment, 710
  García-Gil et al., 2019: Sustainability indicator for the prevention of
potential thermal interferences between groundwater heat pump systems in
urban aquifers. Renewable energy 134

-What is the origin of the thermal parameters showed in Fig. 2? Please, provide the citation.

-Figs. 2 and 3: I think that the quality and the information provided by these two figures should be improved.

Good look!

Reviewer 3 Report

I find it difficult to identify the novelty of this paper. I see no novelty compared to for example the review study, co-authored by Cui (10.1016/j.rser.2018.05.063) or the recent publication, including validation by Laferierre et al (10.1016/j.geothermics.2019.101788).

My key comment is that the authors should pinpoint the novelty more clearly. If the novelty cannot be identified more clearly, then I would reject the paper. At least a major revision is required.

Round 2

Reviewer 1 Report

1) In the Responses, Authors explained that Solar System is only in the actual engineering project and not included in the numerical simulation which is the main part of this article. However, from the paper title "a solar-ground source heat pump system" is perceived as a major theme and concern that research is focused on it. Hence, it is just an advice to remove the term "solar" from the title. 

2) Regarding solar heating, it must be clearly mentioned in the Introduction part also, may be after the objectives, that it is not included in the numerical simulation.

3) It may be better to show figure of GH-12WT09 in the paper itself. And parameters can be briefly described in text.

Reviewer 3 Report

The authors have improved the paper significantly. The novelty is much more clearly described. 

I just have 4 minor comments for improvement:

English language and style check is recommended.

Line 15/16: this sentence may be extended: ..... is established; the level of model integration and validation define the novelty of this paper.

Line 136: This sentence should be rephrased. 

Line 438, Figure 13: Please consider to add scheme 3 to this figure for comparison. 
